# Complex Networks and the *b*-Value Relationship Using the Degree Probability Distribution: The Case of Three Mega-Earthquakes in Chile in the Last Decade

**DOI:** 10.3390/e24030337

**Published:** 2022-02-26

**Authors:** Fernanda Andrea Martín, Denisse Pastén

**Affiliations:** Deparment of Physics, Faculty of Sciences, University of Chile, Santiago 8380453, Chile; denisse.pasten.g@gmail.com

**Keywords:** complex networks, earthquake, characteristic exponent, *b*-value

## Abstract

Studies from complex networks have increased in recent years, and different applications have been utilized in geophysics. Seismicity represents a complex and dynamic system that has open questions related to earthquake occurrence. In this work, we carry out an analysis to understand the physical interpretation of two metrics of complex systems: the slope of the probability distribution of connectivity (γ) and the betweenness centrality (BC). To conduct this study, we use seismic datasets recorded from three large earthquakes that occurred in Chile: the $M_{w}$8.2 Iquique earthquake (2014), the $M_{w}$8.4 Illapel earthquake (2015) and the $M_{w}$8.8 Cauquenes earthquake (2010). We find a linear relationship between the b-value and the γ value, with an interesting finding about the ratio between the b-value and γ that gives a value of ∼0.4. We also explore a possible physical meaning of the BC. As a first result, we find that the behaviour of this metric is not the same for the three large earthquakes, and it seems that this metric is not related to the b-value and coupling of the zone. We present the first results about the physical meaning of metrics from complex networks in seismicity. These first results are promising, and we hope to be able to carry out further analyses to understand the physics that these complex network parameters represent in a seismic system.

## 1. Introduction

Despite the long tradition of studying seismicity, this area still has open questions due to the complexity of the underlying dynamics involved in earthquake occurrence. Two important laws have been established over time: one is the Omori law [1], which is related to aftershocks, and the second is the well-known Gutenberg–Richter law [2], which expresses a relationship between frequency and magnitude. In this sense, active seismic areas of the planet are interesting regions to analyse the behaviour of earthquakes and the complexity involved in their underlying physical processes.

This work is located in South America, where the subduction process occurs between the Nazca tectonic plate and the South American tectonic plate. This strong interaction has caused great earthquakes throughout history. In particular, we are interested in the Chilean coast, which is located in the subduction zone between these two plates, converting Chile into a seismically active zone. Since 1995, Chile has experienced four large earthquakes with magnitudes greater than $M_{w}$8.0 and five large earthquakes with magnitudes greater than $M_{w}$7.0. Seven of those earthquakes were located in the northern zone of Chile. The last decade, from 2010 to the present, has been particularly active, with five of these large earthquakes. In this article, we focus our analysis on three large earthquakes that occurred in Chile: the $M_{w}$8.2 Iquique earthquake in the northern zone of Chile; the $M_{w}$8.4 Illapel earthquake in the north-central zone of Chile and the $M_{w}$8.8 Cauquenes earthquake in the central zone of Chile. Considering the high seismic activity along Chile, we focus our attention on the Gutenberg–Richter law, which shows scale-free behaviour between the frequency and magnitude of seismic events. Many mechanisms have been proposed to account for the observed spatial and temporal variability of *b*, ranging from a fracturing degree and material properties to a stress concentration degree. In particular, Schorlemmer et al., in 2005 [3], proposed a relationship between stress and the *b* value of seismic zones. The *b* value acts as a stress meter that depends inversely on the differential stress. This means that a high value of *b* indicates low stress, and a low value of *b* indicates high stress in the zone. Schurr et al. (2014) [4] analysed the b-value and the interlocking zone of the Iquique rupture, showing the difference between this large earthquake and other mega-earthquakes and concluding that this earthquake broke the zone of lower coupling. Tilmann et al. (2016) [5] conducted a deep analysis of the Illapel earthquake using the b- value, showing that this seismic event had very similar characteristics to the 1943 large earthquake in the same zone. Tassara et al. (2016) [6] explored a new way to analyse a megathrust by combining and analysing the afterslip and b-value, finding a distribution of the b-value after the occurrence of this large earthquake. Thus, we have studies on the b-value, but deep and systematic studies have also been carried out on coupling along the Chilean coast. Métois et al. (2012, 2013a, 2013b and 2016) [7,8,9,10] dedicated a series of studies to the study of the Chilean coast coupling, providing a detailed description of the coupling values throughout Chile.

On the other hand, complex networks have been shown to be a strong tool to understand the complexity inside real systems. From a formal point of view, a complex network is a set of nodes or vertices connected via axes. A complex network has certain nontrivial statistical and topological properties that do not occur in simple networks. Research in this area has undergone important development in recent years, highlighting different fields, such as biology [11,12,13], communication [14] and social relations [15]. In particular, the application of complex networks to geophysics has been helpful in finding behaviours such as the power law of probability distributions, small-world networks, fractals or the reversibility of the time series. Considering the characteristics of a complex network, earthquakes can be studied as a visibility graph [16,17], considering the time evolution of the magnitudes [18,19] or as a spatiotemporal distribution [20].

In this study, we carry out an initial approach to determine a possible relationship between the critical exponents of the complex network theory and the physical parameters of earthquakes, such as the *b*-value of the Gutenberg–Richter law. We analyse three seismic datasets measured for three large earthquakes that occurred in Chile as a directed earthquake complex network based on the time sequence of the seismic event occurrence using the Abe–Suzuki method [18,19,20,21,22]. The article is organized as follows: in Section 2, we give details related to the seismic datasets measured for the three large earthquakes, how to build the complex network using seismic data events and the method to compute the b-value, and we explain the maximum likelihood estimation method. In Section 3, we show the results and provide a discussion. Finally, in Section 4, we provide the conclusions.

## 2. Data and Network Analysis

### 2.1. Seismic Data Sets

The seismic data were measured by the National Seismological Center of Chile (Centro Sismológico Nacional, CSN) [23] between January 2005 and March 2017, containing 12 years of measurements with 38,083 seismic events in a zone between $17.9^{\circ }$ and $39.1^{\circ }$ South Latitude and between $67.5^{\circ }$ and $75^{\circ }$ West Longitude, with a maximum depth of 200.0 km. Therefore, we analyse the *b*-value, the critical exponent γ and the betweenness centrality for the three large earthquakes greater than $M_{w}$8.0 that occurred in Chile in 2010, 2014 and 2015. From the initial large dataset measured between 2005 and 2017, we study the seismic events closer to the rupture zone of the main earthquakes, i.e., we consider the rectangle formed between the latitude coordinates that includes the longitude of the rupture zone for each earthquake, and in longitude coordinates, we use the complete seismic events that occurred in each zone. This decision is based on the amount of data collected in the original data; in this way, we have a large amount of data for the analysis. Figure 1 shows the rectangles of seismic events used in this analysis.

The data were collected by the National Seismological Center (Centro Sismológico Nacional) [23] in the format of date, hypocentre and magnitude. For the present study, the hypocentre is shown in kilometres. The latitude is represented by the angle θ, and the longitude is represented by the angle ϕ. Latitude and longitude are converted into kilometres using the following expressions:$$d_{i}^{NS}=R\left(\theta_{i}-\theta_{0}\right),$$
$$d_{i}^{EW}=R\left(\phi_{i}-\phi_{0}\right)\cos\left(\theta_{av}\right),$$
$$d_{i}^{z}=z_{i},$$
where $z_{i}$ is the depth, and $\theta_{AV}$ is the average latitude. $\theta_{0}$ and $\phi_{0}$ are the minimum values for the latitude and longitude, and *R* is the radius of the Earth—6370 km in this study.

The entire seismic dataset contains seismic events of great magnitude, of which three of them stand out. The first one is the Cauquenes earthquake on 27 February 2010, with a magnitude of $M_{w}=8.8$ on the Richter scale. The dataset used in this analysis contains 14,230 seismic events between January 2005 and March 2017, and these events are along and close to the rupture zone, which was 500 km by 140 km, a very large segment in central Chile [24]. The rectangle used for this zone is between 33.5° and 37.7° South Latitude and 68.232° and 74.974° West Longitude. The hypocentre was located in the geographic coordinates 73.239° West Longitude and 36.290° South Latitude, with an estimated depth of 30 km, according to the National Seismological Center [25].

The second event was the Iquique earthquake of 1 April 2014, with a magnitude of $M_{w}=8.2$ on the Richter scale. The dataset contains 8362 seismic events between January 2005 and March 2017. The rupture zone was approximately 100 km by 50 km in a seismic gap zone in northern Chile [26]. The rectangle used for this zone is between 19.0° and 21.0° South Latitude and 68.001° and 72.104° West Longitude. The hypocentre coordinates are 19.572° South Latitude and 70.908° West Longitude, with a depth of 38.9 km, according to the National Seismological Center [27].

The last earthquake was the Illapel earthquake on 16 September 2015, with a magnitude of $M_{w}=8.4$ on the Richter scale. The dataset contains 9746 seismic events between January 2005 and March 2017. The rupture zone was approximately 50 km to the north and 75 km to the south of the hypocentre [5]. The rectangle used for this zone is between 30.0° and 32.2° South Latitude and 68.008° and 72.991° West Longitude. The location of the hypocentre has the coordinates 31.637° South Latitude and 71.741° West Longitude, with a depth of 23 km, according to the National Seismological Center [28].

The maximum depth for the three datasets is 200 km.

### 2.2. Gutenberg–Richter Law

Distributions of earthquakes in any region of the Earth typically satisfy the Gutenberg–Richter [2] relationship given by
(1)$$log_{10}\left(N\right)=a-bM,$$
where *N* is the cumulative number of earthquakes greater than magnitude *M*. The values *a* and *b* indicate the intercept and the slope of the power law. The most important parameter in this case is *b*; a higher value of *b* indicates a larger proportion of small earthquakes, and a lower value refers to a smaller proportion of small earthquakes [2].

The method that we use to estimate the threshold magnitude is the maximum curvature technique (MAXC) proposed by Wiemer and Wyss (2000) [29]. The maximum curvature technique computes the maximum value of the first derivative of the frequency–magnitude curve. This matches the magnitude with the highest frequency of events in the noncumulative frequency magnitude distribution (FMD) in practice.

### 2.3. Seismic Complex Network

To perform the time-based complex network analysis, we build the network for each region, as shown in Figure 1, and each network contains the rupture zone of the Cauquenes earthquake, Illapel earthquake and Iquique earthquake. A complex network consists of nodes connected between them through links. To build the complex network with the seismic datasets, we divide each zone into cubic cells with a side size of Δ=10 km. Afterwards, we check if a hypocentre is inside the cubic cell; if so, the cell is called a node. Then, we place the connections between nodes following the temporal sequence of the seismic events. The direction of the connections between nodes is defined through the temporal sequence of the seismic events in the region, as shown in Figure 2. Therefore, we have a cubic cell that contains hypocentres as nodes and connections between nodes following the time occurrence of the seismic events. Thus, we built a directed complex network using the method of Abe–Suzuki [18].

After building the complex network with seismic events, we compute the probability distribution of the degree of connectivity of the nodes. Typically, this distribution shows the complexity of the system through exponential, Gaussian or scale-free behaviour. In the case of time-based complex networks built with seismic datasets, the probability distribution of the connectivity gives a power law of [18,21].
$$P\left(k\right)=k^{-\gamma },$$
where γ is the characteristic exponent of the scale-free distribution.

We also calculate the cumulative distribution of the betweenness centrality (BC). This metric is a measure of how important node *m* is in the shortest distance connections between all other pairs of nodes. Hence, if the BC is large, the nodes are more active. To calculate the betweenness centrality of the network, we use the definition given by Freeman (1977) [30], which tells us that the BC of node *j* is
$$g\left(j\right)=\sum_{i\neq j\neq k}\frac{d_{ik}\left(j\right)}{d_{ik}},$$
where $d_{ik}$ is the number of geodesics between nodes *i* and *k*, and $d_{ik}\left(j\right)$ is the number of geodesics between *i* and *k* that contain node *j* as the central node. Usually, the normalization of the BC is performed by dividing by the number of pairs of nodes, excluding *j*, and dividing by (N−1)(N−2) for the directed network. However, as we have a large number of nodes and not all nodes have connections between them, another type of normalization will be used that allows us to position the BC in a range of [0,1]. This normalization is given as
$$normal\left(g\left(j\right)\right)=\frac{g(j)-min(g)}{max(g)-min(g)},$$
where max(g) and min(g) are the maximum and minimum BC of the network, respectively.

The cumulative distribution of BC follows a power law if
$$B\left(x<g\right)∼g^{-\delta }.$$
In the three zones, our analysis consists of calculating the characteristic exponent γ for the scale-free connectivity distribution of the network, the slope for the cumulative distribution of the betweenness centrality and the value of BC and δ.

### 2.4. Maximum Likelihood Estimation and Linear Regression

To carry out the proposed analysis, we need to compute the slope of several power law distributions: the Gutenberg–Richter law, the probability distribution of the connectivity and the cumulative probability distribution of the betweenness centrality. To obtain an estimation of the power law index, we apply two strategies, namely, maximum likelihood estimation (MLE) and linear regression (LR).

For LR, we apply a simple linear regression analysis in log–log scale of the histogram obtained from the connectivity of the betweenness centrality distribution of the complex network constructed for each of the three datasets. For MLE, we used the approach proposed by Goldstein et al. (2004) [31], where the range of applicability of the estimated scale-free distribution is found from the Kolmogorov–Smirnov type of test proposed by Goldstein et al. (2004) [31].

## 3. Results and Discussion

First, to conduct a reliable analysis, we compute the magnitude of completeness. The first column in Figure 3a–c shows the magnitude of completeness for the three large earthquakes calculated using the maximum curvature technique (MAXC) [29]. The value is $M_{c}$3.0 in the case of the Cauquenes earthquake (2010), with 9560 seismic events, $M_{c}$2.9 for the Iquique earthquake (2014), with 5904 seismic events, and $M_{c}$3.0 for the Illapel earthquake (2015), with 7316 seismic events. The quality of the dataset is very similar, and we can compare the results of these three datasets. Therefore, the *b*-value is computed based on the magnitude of completeness. Table 1 shows the *b*-value for each zone. We can observe that the highest value is found for the Cauquenes earthquake, with b=0.964±0.006, while for the Illapel earthquake, we find b=0.949±0.004, and the lowest value is for the Iquique earthquake, with b=0.758±0.004. From Métois et al. (2013), we know that the Iquique earthquake ruptured a zone with a low coupling, but following the value of *b*, this zone presents high stress from the conclusions of Schorlemer et al. (2005). For the Illapel earthquake, Métois et al. (2012) and Tilmann et al. (2016) agreed on the low coupling in the northern zone of the Metropolitan segment, but the value of *b* is high, showing a low level of stress [3,32]. In the case of the Cauquenes earthquake, the value of *b* is high, showing low stress [3,32], and the coupling is also high. Of course, we know that a seismic system is much more complex and that other metrics are involved in this process, but for this analysis, we focus on coupling and the *b*-value.

With the datasets filtered, we build the seismic complex network using the magnitudes of completeness, and we characterize each complex network using the degree distribution, with the critical exponent γ, and the cumulative distribution of the betweenness centrality, with the critical exponent δ, as explained in Section 2.

The second column in Figure 3a–c shows scale-free behaviour for the probability distribution of the connectivity and the degree of the nodes for the three mega-earthquakes. It is possible to observe how both methods—maximum likelihood estimation (MLE) and linear regression (LR)—give similar results for this metric. The values of the slopes are shown in Table 1. For the second column in Figure 3, the Kolmogorov–Smirnov test gives the ranges $10^{0.4}\leq k\leq 10^{1.6}$ for the Cauquenes earthquake, $10^{0.4}\leq k\leq 10^{1.9}$ for the Iquique earthquake and $10^{0.4}\leq k\leq 10^{1.9}$ for the Illapel earthquake. We find the greatest value of γ for the Illapel earthquake and the lowest value for the Iquique earthquake, similar to the previous results obtained for the *b*-value.

The third column in Figure 3a–c shows scale-free behaviour in the cumulative distribution of the betweenness centrality for the three large earthquakes. Table 1 shows the values of the slope in the case of the exponent δ for the BC. The applicability range with the Kolmogorov–Smirnov test for this metric is $10^{-2.9}\leq k\leq 10^{-0.3}$ for the Cauquenes earthquake, $10^{-2.4}\leq k\leq 10^{-0.1}$ for the Iquique earthquake and $10^{-2.5}\leq k\leq 10^{0.4}$ for the Illapel earthquake. In this case, MLE gives a better fit than LR. Considering the MLE values for the δ exponent, we find the same relationship as above: the highest value corresponds to the Cauquenes earthquake, and the lowest value of critical exponent δ corresponds to the Iquique earthquake. The three datasets analysed show the same behaviour: with a high value of *b*, there is a high value of γ and δ. Therefore, it seems that there is a directed proportional relationship between the *b*-value and these two critical exponents of complex networks. We conduct a simple test between the value of γ and the *b*-value, and we find a simple relationship between these two parameters. If we divide $\frac{b}{\gamma }$ for the Iquique earthquake, we find 0.40; for the Illapel earthquake, we find 0.45; and for the Cauquenes earthquake, we find 0.42. It seems that a constant of proportionality appears for this ratio.

To analyse the ratio between the *b*-value and the critical exponent γ, we shuffle the time series to test the reliability of the results. After shuffling ten times the seismic datasets for each zone, we find a stable value of the ratio, keeping the value obtained for the datasets without shuffling. The results are shown in Table 2.

Additionally, we calculate the ratio before and after the main earthquake. The results are shown in Table 3 and Table 4. We observe fewer seismic events before the main earthquake in the cases of Cauquenes and Iquique, as long as the amount of data before the main shock for the Illapel earthquake is greater than the seismic events afterwards.

The Kolmogorov–Smirnov test for the values of γ before the main shock gives a range of $10^{0.2}\leq k\leq 10^{2.0}$ for the Cauquenes earthquake, $10^{0.2}\leq k\leq 10^{2.2}$ for the Iquique earthquake and $10^{0.2}\leq k\leq 10^{2.2}$ for the Illapel earthquake. The range of the Kolmogorov–Smirnov test after the principal earthquake is $10^{0.4}\leq k\leq 10^{1.6}$ for the Cauquenes earthquake, $10^{0.2}\leq k\leq 10^{1.9}$ for the Iquique earthquake and $10^{0.2}\leq k\leq 10^{1.9}$ for the Illapel earthquake.

We find a small change in the values of the ratio $\frac{b}{\gamma }$ before and after the main shock. For the datasets before and after the large earthquakes, we find a value of the ratio between 0.35 and 0.45, which is very close to the value of 0.4 found in Table 1. The Cauquenes earthquake was a subduction seismic event without high precursors, and the *b*-value for this zone was high, which could be related to high stress. The zone of the Cauquenes earthquake rupture had high coupling before the main shock, and for this large earthquake, the value of the ratio $\frac{b}{\gamma }$ before the main event was 0.35, while after the main shock, this ratio increased to 0.39, close to 4.0. Additionally, the Iquique earthquake was an interplate subduction earthquake similar to the Cauquenes and Illapel earthquakes, but the Iquique earthquake was different because the foreshocks and aftershocks were located on the interplate interface but at shallow depths. The value of the ratio before the main shock was higher than that of the other earthquakes — $\frac{b}{\gamma }=$0.43. Before $M_{w}$8.2, this zone was considered to have low background activity by some researchers, such as Scholz (1998) [33], but there was a large foreshock of magnitude $M_{w}$6.7 on 16 March 2014, 2 weeks before the large earthquake, while this ratio decreased to 0.36 after the main shock. Finally, the Illapel earthquake occurred between two high coupling zones [7,8] and near the northern zone of the rupture of a mega-earthquake that occurred in 1730 ($M_{w}∼$9.0); therefore, the zone ruptured in 2015 was a seismic gap. In this case, the value of the ratio $\frac{b}{\gamma }$ was 0.40 before the main shock, and this value increased after this event to a value of 0.45, following the same behaviour as the Cauquenes earthquake.

The Cauquenes and Illapel earthquakes did not have an increase in seismic activity before the large earthquakes, which may be the reason why the values of these ratios increased after the main shock. In addition, the Iquique earthquake had foreshocks two weeks before the main seismic event, and in this case, the value of this ratio increased after the principal seismic event.

Figure 4 shows the nodes with the highest values of BC for each earthquake. This metric exhibits the most important nodes following the shortest paths between nodes in the network and the earthquake of the zone. The fourth column shows the BC values normalized; therefore, the highest value that they can have is 1; therefore, we are taking into account that a high BC is over 0.3 since most nodes have values less than 0.1. The fifth column shows the original values of BC without normalization. Table 5, Table 6 and Table 7 show the location of the centre of the nodes with high BC values with associated numbering. These values are shown for each large earthquake, and the tables include the BC value for the three zones, which means that we could have more than one node in the marked centre. These tables show some nodes with a BC equal to 1.0. These results are due to the normalization used and do not reflect the number of shortest paths between two nodes divided by the number of the shortest paths passing through node *j*. The original BC values are very low for the three cases.

We observe from Figure 4 that the epicentre of the Cauquenes earthquake was not close to the hot points with a high BC, but most seismic events occurred in those three hot points, and these events corresponded to seismicity that occurred in 2010 after the occurrence of the megathrust. In the case of the Iquique earthquake, we can observe two marked zones: a zone to the south of the epicentre of the main event with two hot points with high BC values where only four seismic events occurred before the large earthquake, and most of this seismicity occurred after the main earthquake in 2014. We observe another zone close to 69.0° West Longitude. These six hot points with high BC values were not related to the Iquique earthquake. This zone contained only 14 seismic events after the Iquique earthquake in 2014. That zone contains the Tarapaca earthquake $M_{w}$7.9 occurring on 13 June 2005 and the $M_{w}$6.3 foreshock of the Iquique earthquake on 24 March 2008. The BC map of the Illapel earthquake shows two marked zones similar to the Iquique earthquake. The first zone, in this case, is between 30.4° and 30.9° South Latitude and contains five hot points with high BC; only eight seismic events occurred before the large earthquake of Illapel; therefore, these hot points were related to the main earthquake in 2015. The second zone is close to the epicentre of the Illapel earthquake. The seismicity at these three hot points is related to previous seismic activity in that zone and to activity as a result of the earthquake. Therefore, for the Cauquenes earthquake, the central nodes move toward the north and are related to the megathrust; for the Iquique earthquake, only the nodes in the south were related to the main shock; and for the Illapel earthquake, there are two zones, and both were related to the main earthquake.

## 4. Conclusions

We analysed three large earthquakes measured in Chile to understand if there is a relationship between the complex network theory and the physical processes that acted in the earthquake occurrences. We compute the values of the magnitude of completeness for each seismic zone studied: $M_{w}$3.0 for the Cauquenes earthquake, $M_{w}$2.9 for the Iquique earthquake and $M_{w}$3.0 for the Illapel earthquake. The study on the *b*-value gives different values for the three seismic datasets: b= 0.964 for the Cauquenes earthquake, b=0.758 for the Iquique earthquake and b=0.949 for the Illapel earthquake. These results suggest greater stress for the Iquique earthquake zone and a low value of stress for the Cauquenes and Illapel earthquakes. After this first seismic analysis, we carried out a complex network study to find the scale-free behaviour for the probability distribution of connectivity and for the cumulative distribution of the betweenness centrality. From complex networks, we find different values of the slope γ for the probability distribution of the degree. We use two methods to compute this slope, MLE and LR, in the case of the critical exponent γ, and the best results are from MLE. We obtain γ= 2.3 for the Cauquenes earthquake, γ= 1.8 for the Iquique earthquake and γ= 2.1 for the Illapel earthquake. These values show the same behaviour as the *b*-value; the greatest value is for the Cauquenes earthquake, and the lowest value is for the Iquique earthquake; therefore, it seems that there is a linear relationship between these two parameters. If we calculate the ratio $\frac{b}{\gamma }$ for each seismic zone, we find a particular number close to 0.4. These results suggest a linear relationship between the physical parameter *b* for the magnitude distribution of earthquakes and γ, a critical exponent from the complex network theory. This indicates that for a zone with low stress (high value of *b*), we also obtain a high value of the slope for the probability distribution of connectivity; therefore, we have a less connected zone. In addition, if we observe a low value of *b* (less stress), we obtain a small value of γ; therefore, we have a more connected zone. In the analysis of these parameters before and after the main earthquakes, we observe stressed areas before each large earthquake with a slightly connected complex network, and after the main seismic event, these zones show a decrease in stress and an increase in the connectivity of the complex network. This could be the first step in understanding how complex networks and the physics of an earthquake occurrence are related.

Finally, we performed an analysis related to the importance of the nodes in the network. We computed a normalized value of the BC for the nodes in each zone studied, which revealed that the hot points for each large earthquake show high seismic activity related to the main earthquakes, even when the hot points are very far from the epicentre of the megathrusts. These points with high BC values contain mostly seismic events that occurred immediately after the main shock. These results suggest that this BC method could be helpful for analysing the activity produced by a large earthquake and its progress in time.

The main results suggest a direct proportionality between the *b*-value, considered as a measure of the stress in a seismic zone, and the value of the critical exponent γ obtained from the probability distribution of the degree of the nodes. We are working on a depth analysis of these results to understand the role that the BC could play in the study of aftershocks. 

## Figures and Tables

**Figure 1 entropy-24-00337-f001:**
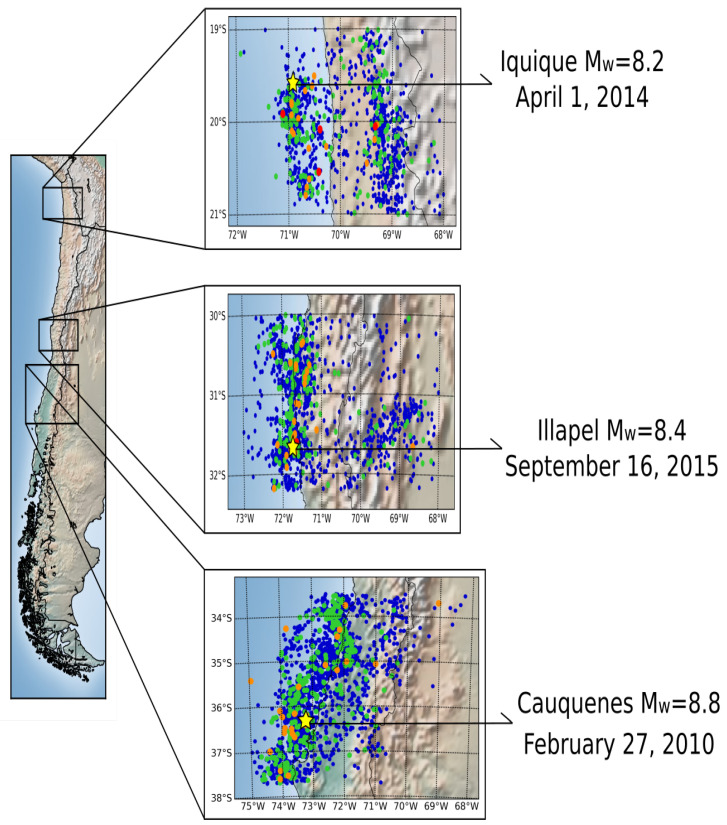
Map with the seismic events with magnitudes greater than $M_{w}=4.0$, where the blue dots represent values of magnitude between $4.0<M_{w}\leq 5.0$, the green dots represent values of magnitude between $5.0<M_{w}\leq 6.0$, the orange dots represent values of magnitude between $6.0<M_{w}\leq 7.0$ and the red dots represent values of magnitude between $7.0<M_{w}\leq 8.0$. The star indicates the epicentre of the earthquakes with magnitudes greater than $M_{w}=8.0$.

**Figure 2 entropy-24-00337-f002:**
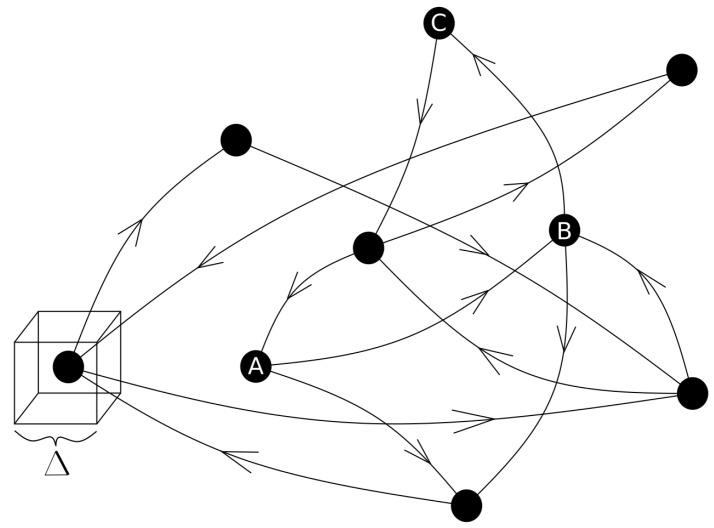
The earthquake network: a directed network where the black dots represent the nodes as the center of the cubic cell in which the seismic events occurred. The side size of the cell is Δ=10 km. The connections (edges) between nodes follow the temporal sequence of the seismic events. In this schema, the first event in time is node *A*, the second is node *B* and the third is node *C*.

**Figure 3 entropy-24-00337-f003:**
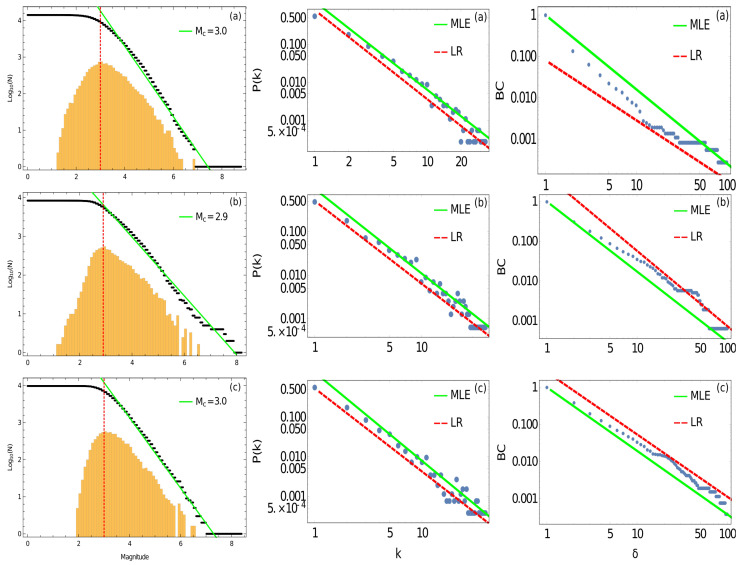
The three zones studied: (**a**) Cauquenes earthquake 2010; (**b**) Iquique earthquake 2014; and (**c**) Illapel earthquake 2015. The first column shows the Gutenberg–Richter law for each earthquake, where the slope was calculated using the complete magnitude of each zone, as indicated in the legend. The red points show the maximum curvature technique (MAXC). The second column is the distribution of the probability of connectivity and the corresponding slope calculated using the maximum likelihood estimation (MLE) and linear regression (LR). The third column shows the cumulative distribution of betweenness centrality calculated using the maximum likelihood estimation (MLE) and linear regression (LR).

**Figure 4 entropy-24-00337-f004:**
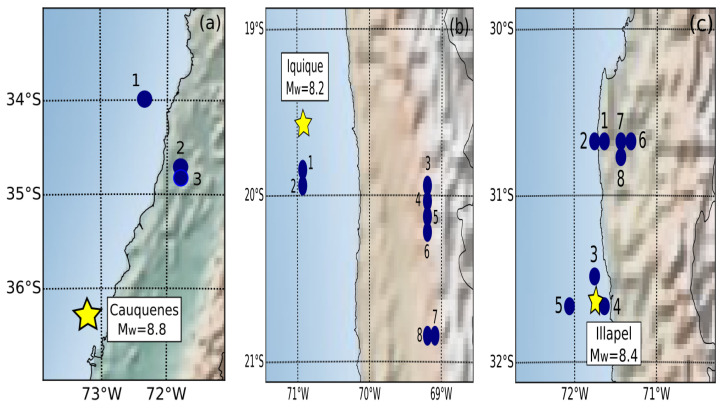
Location of the node centres with high betweenness centrality (BC) values in blue dots, indicating the corresponding epicentre of the main earthquake in each zone with a yellow star: (**a**) Cauquenes earthquake; (**b**) Iquique earthquake; (**c**) Illapel earthquake.

**Table 1 entropy-24-00337-t001:** Table with the values of *b*, the Gutenberg–Richter law exponent, the value of γ, the characteristic exponent of probability distribution of connectivity and the value of the δ exponent of the distribution of betweenness centrality of each earthquake. The exponents δ and γ were calculated using the maximum likelihood method (MLE) and a simple linear regression (LR), and the value of *b* was calculated using only linear regression (LR).

Zone	$M_{c}$	b	γLR	γMLE	δ LR	δMLE
Cauquenes	3.0	0.964±0.006	2.3±0.1	2.3±0.1	1.44±0.04	1.79±0.02
Iquique	2.9	0.758±0.004	1.9±0.1	1.9±0.1	1.89±0.05	1.76±0.03
Illapel	3.0	0.949±0.004	2.1±0.1	2.2±0.1	1.68±0.02	1.71±0.01

**Table 2 entropy-24-00337-t002:** Results of the *b*-value and the critical exponent γ for the average value of ten shuffles of the seismic datasets.

Zone	Number of Seismic Events	*b*-Value	Average γ	$\frac{b}{\gamma }$
Cauquenes	9560	0.964 ± 0.006	2.3 ± 0.1	0.42
Iquique	5904	0.758 ± 0.004	1.9 ± 0.1	0.40
Illapel	7316	0.949 ± 0.004	2.1 ± 0.1	0.45

**Table 3 entropy-24-00337-t003:** Results of the *b*-value and the critical exponent γ before the main earthquake.

Zone	$M_{c}$	Seismic Events	*b*-Value	γ LR	γ MLE	$\frac{b}{\gamma }$
Cauquenes	3.0	1503	1.06 ± 0.01	3.0 ± 0.2	3.1 ± 0.1	0.35
Iquique	3.0	2620	0.91 ± 0.01	2.1 ± 0.1	2.2 ± 0.1	0.43
Illapel	3.0	3970	1.06 ± 0.01	2.6 ± 0.1	2.6 ± 0.1	0.40

**Table 4 entropy-24-00337-t004:** Results of the *b*-value and the critical exponent γ after the main earthquake.

Zone	$M_{c}$	Seismic Events	*b*-Value	γ LR	γ MLE	$\frac{b}{\gamma }$
Cauquenes	3.0	8057	0.958 ± 0.006	2.4 ± 0.1	2.4 ± 0.1	0.39
Iquique	2.8	3337	0.728 ± 0.004	2.0 ± 0.1	2.0 ± 0.1	0.36
Illapel	3.2	2784	0.861 ± 0.004	1.9 ± 0.1	2.0 ± 0.1	0.45

**Table 5 entropy-24-00337-t005:** Table with the location of the hypocentre of the nodes with a high betweenness centrality (BC) in the zone of the Cauquenes earthquake.

Number	Node	Events	BCN	BC	Latitude	Longitude	Depth [km]
1	9089	70	0.2862	786,930.7	−33.995	−72.341	35
2	9580	80	0.5798	1,594,251.7	−34.714	−71.793	35
	12,494	98	0.7835	2,154,402.5			45
3	9642	38	0.2506	689,164.0	−34.804	−71.793	35
	12,556	116	1	2,749,813.5			45

**Table 6 entropy-24-00337-t006:** Table with the location of the hypocentre of the nodes with a high betweenness centrality (BC) in the zone of the Iquique earthquake.

Number	Node	Events	BCN	BC	Latitude	Longitude	Depth [km]
1	3384	72	0.4740	155,753.5	−19.85	−70.92	35
2	3427	66	0.4717	155,006.2	−19.94	−70.92	35
3	9343	56	0.6171	202,770.3	−19.94	−69.20	95
4	9386	50	0.5239	172,136.9	−20.03	−69.20	95
5	9429	43	0.4574	150,313.8	−20.12	−69.20	95
6	9472	45	0.4954	162,788.8	−20.21	−69.20	95
7	10,761	77	1	328,592.8	−20.84	−69.10	105
8	10,762	56	0.6161	202,430.9	−20.84	−69.20	105

**Table 7 entropy-24-00337-t007:** Table with the location of the hypocentre of the nodes with a high betweenness centrality (BC) in the zone of the Illapel earthquake.

Number	Node	Events	BCN	BC	Latitude	Longitude	Depth [km]
1	3970	75	0.9482	652,641.5	−30.67	−71.64	35
2	3971	54	0.7370	507,235.2	−30.67	−71.74	35
3	4403	44	0.6246	429,913.9	−31.48	−71.74	35
4	4498	36	0.4413	303,757.7	−31.66	−71.64	35
5	4502	39	0.4601	316,677.4	−31.66	−72.06	35
6	5167	33	0.4209	289,678.1	−30.67	−71.32	45
7	5168	57	0.8185	563,344.6	−30.67	−71.43	45
8	5216	83	1	688,272.1	−30.76	−71.43	45

## Data Availability

The datasets used or analysed during the current study are available at the following link: https://datos.uchile.cl/dataset.xhtml?persistentId=doi:10.34691/FK2/VENURN (accessed on 23 January 2020).

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
