# Peer review of "Complex Networks and the b-Value Relationship Using the Degree Probability Distribution: The Case of Three Mega-Earthquakes in Chile in the Last Decade"

_entropy, 2022, doi:10.3390/e24030337_

Round 1
Reviewer 1 Report
The article “Complex networks and b-value relation through the degree probability distribution: the case of three mega-earthquakes in Chile in the last decade.” by Martin and Pasten deals with a very interesting topic. Although one of the authors, Prof. Pasten is an expert in complex networks and for sure the application of complex networks to seismicity is an interesting and promising field, a number of unclear points indicate that this article should considerably be improved before it can be accepted for publication. Specifically:
- It is not clear why the article focuses to the three mentioned M>8 earthquakes, while the analyzed data extend several years before as well as after each one of these earthquakes. Maybe the authors are not focusing on the mentioned three events but instead on the pointed out three seismogenic regions? But then, what would be the interpretation of the results?
- To the extent possible from the visual inspection of Figure 1, it seems that at least in the case of two of them (Iquique and Illapel) events of magnitude between 7.0 and 8.0 have been included in the analysis. Even though no specific information about the time of occurrence of these events is given, one would expect that the specific events are of comparable importance with the earthquakes in focus. Given that the authors focus on the three mentioned M>8 earthquakes, one would expect the authors to separately study the pre-earthquake time period of each one of the earthquakes in focus, for example from the time that the previous ~M7 earthquake’s aftershock sequence is finished up to the time of occurrence of the M>8 event of interest, and separately study the aftershock period of the M>8 event of interest.
- The phenomenological connection between b and gamma for the three presented cases is a kind of universally standing relation? Is the calculated ration value characteristic of the studied region? Maybe much more investigation (for many more cases) should be done before one can end up with a solid interpretation.
- Are the authors sure that the results presented in Figure 4 are related (or mainly connected) to the specific three M>8 earthquakes? Could these be related to other significant event(s) of the same area that happened during the studied time period? Would it be an explanation of the distance between the earthquakes in focus and the revealed nodes with a high betwenness centrality? Would the picture be different if one studies other time periods or even sub-periods of the considered ones for the same regions?
Some minor issues:
- The non-cumulative frequency-magnitude distribution is shown in Figure’s 3 left column (red dots) but this not clarified in the figure caption or the text.
- At some places the authors use the term “complete magnitude” instead of the correct one “magnitude of completeness”.
Reviewer 2 Report
The authors analyzed seismic catalogs (January 2005 to March 2017) of three rapture zones in Chile where a large earthquake occurred in each of these zones. More specifically, they found the magnitude threshold of each of the zones (which is about 3), the Gutenberg-Richter exponent, the exponent of the degree of the nodes, and the betweenness-centrality exponent. Generally speaking, they found different values for the different regions and attempted to find an empirical relation between the Gutenberg-Richter exponent and the exponent of the degree nodes to be around 0.4.
Unfortunately, the manuscript contains many typographical and grammar mistakes which make the paper difficult to read. These must be fixed. In addition, it is not clear to me what is the added value of the present analysis. Indeed, the authors attempted to draw some conclusions, but these were vague and unclear. I also believe that their conclusion regarding the ratio between the Gutenberg-Richter exponent and the exponent of the degree nodes is wrong. Thus, I suggest rejecting the paper at this stage and to consider it for publication only after substantial revisions of both the analysis and physics. Below I provide some additional comments/suggestions.
* Basically the authors concentrate on specific regions and not on specific earthquakes. This should be clarified and the title, abstract, and text should be modified accordingly.
* According to what criteria the lateral extent of the different regions was chosen? Is it according to the rapture zone extent? Can you add a map of the different rapture zones? Why consider only these regions? I believe that it will make sense to analyze the entire seismic catalog of Chile.
* Please elaborate on Abe and Suzuki method.
* As acknowledged by the authors, the normalization of g may be misleading. Can you add also the original values before normalization and interpret them accordingly?
* I believe that the completeness magnitudes should be higher in all three cases as the exponential region extends to the left to the low magnitude crossover. I'm not sure how this would affect the results and analysis. In, addition, logarithmic binning in the two left columns of Fig. 3 may improve the estimation of the exponents--now the higher x-axis values dominant the results (at least those of the linear regression).
* The ratio b/\gamma~0.4 is not clear to me. Under shuffling of the catalogs, b should remain the same while \gamma should be changed drastically. So, one should not expect such a relation to be generally valid. Please perform the shuffling analysis to verify the above.
